# Sleep Characteristics in Individuals with Ehlers-Danlos Syndrome

**DOI:** 10.3390/medsci13030085

**Published:** 2025-06-27

**Authors:** Caitlin Crews-Stowe, Frank Tudini, Min-Kyung Jung, Jake Forman, Bernadette Riley, Stephanie Eton, David Levine

**Affiliations:** 1Health and Human Performance Department, University of Tennessee at Chattanooga, Chattanooga, TN 37403, USA; caitlin-crews-stowe@utc.edu; 2Physical Therapy Department, University of Tennessee at Chattanooga, Chattanooga, TN 37403, USAdavid-levine@utc.edu (D.L.); 3College of Osteopathic Medicine, New York Institute of Technology, Old Westbury, NY 11568, USA

**Keywords:** Ehlers–Danlos syndrome, sleep duration, sleep latency, sleeping habits

## Abstract

**Background/Objectives:** The presence of Ehlers–Danlos Syndromes (EDSs) has significant effects on overall health and results in varying levels of pain and disability. The effects of sleep are not well documented in this population. The purpose of this study is to report the sleep characteristics of people with EDS. **Methods:** An electronic survey regarding sleep characteristics was created and distributed through the EDS website. **Results:** Sleep disturbance is common in people with EDS, with 65.3% of respondents sleeping fewer than 8 h and 26.2% averaging fewer than 6 h. Those who slept fewer than 6 h reported more days of poor mental and physical health days. Sleep aids were commonly used with 41.40% of patients regularly taking prescription medication to get to sleep. Sleep latency of greater than 30 min was also found in 67.5% of subjects. **Conclusions:** The results demonstrate an association between people with EDS and poorer sleep duration, increased sleep latency, and increased use of sleep aids including prescription sleep medication compared to the general population. While more research needs to be completed in this area, sleep may be an important aspect to address in the management of EDS.

## 1. Introduction

Ehlers–Danlos Syndromes (EDSs) are a group of genetically heterogeneous connective tissue disorders that vary in how they affect the body and in their genetic causes. There are 13 subtypes of EDS with specific genetic mutations and inclusion criteria required for the diagnosis of each type. Expressions of EDS are known to affect every organ system in the body, most commonly the integumentary, blood vessel, and muscle systems [1]. Characteristics of EDS include joint hypermobility (joints that stretch further than normal), skin hyperextensibility (skin that can be stretched further than normal), and tissue fragility [1,2]. It is also common to see associations with pain, heart disease, sleep disorders, psychiatric disorders, and skin disease in people with EDS [2].

The prevalence of EDS is estimated to be as high as 1 in 5000 individuals, with hypermobile-type Ehlers-Danlos Syndrome (hEDS) comprising 80–90% of all cases [3]. Common functional problems seen in individuals with hypermobility conditions such as hEDS include chronic pain, macro-trauma to joints and soft tissue, disturbed proprioception, bladder and pelvic dysfunction, and gastrointestinal disorders [4]. These problems may result in varying levels of disability that may impact their quality of life (QoL), including sleep. While previous studies have examined the psychological burden of EDS, there is limited research regarding sleep characteristics and sleep quality in individuals with EDS [5]. The Centers for Disease Control and Prevention recognizes sleep as essential for emotional, mental, and physical well-being and recommends 7–9 h of sleep each night [6]. Sleeping less than 7 h per night regularly has been associated with adverse health outcomes [7,8]. It is estimated that 50–70 million Americans suffer from sleep dysfunctions, impacting the physiological systems including immune function, weight gain and obesity, tissue healing, pain modulation, cardiovascular health, cognitive function, learning and memory, and increased risk of death [7,8]. Sleep can also influence an individual’s QoL, with both sleep duration and sleep quality playing pivotal roles. The prior literature indicates that poor sleep is associated with lower QoL scores, particularly affecting physical and mental health domains. Lee and colleagues found in a study involving over 225 thousand adults that individuals reporting poor sleep quality had significantly lower QoL scores compared to those with good sleep quality. Furthermore, poor sleep quality was linked to increased likelihood of experiencing problems with physical activity, pain, and anxiety and/or depression [9,10].

Individuals with EDS have displayed patterns of sleep dysfunction, with Moss et al. reporting poor sleep quality in 61% of patients (*n* = 252) [11]. Sleep dysfunction in EDS can be attributed in part to pain, which leads to impaired sleep quality and fatigue. Persistent pain and overwhelming fatigue are interrelated and common symptoms in chronic conditions like EDS. A recent longitudinal study demonstrated that over time, EDS patients experience statistically significant worsening of pain severity and interference [12,13]. Poor sleep then further amplifies symptoms, creating a vicious cycle that gradually diminishes patients’ capacity for self-care, engagement in daily life activities, and functional status. Ultimately, pain, fatigue, and sleep dysfunction in EDS patients substantially reduces overall quality of life and health, particularly if pain severity is amplified [14,15]. In a study of 273 participants with EDS, 77% suffered from severe fatigue, defined as an overwhelming sense of tiredness, lack of energy, and feeling of exhaustion. The severely fatigued individuals had statistically significantly higher reports of sleep disturbance, concentration problems, pain, and psychological distress [12,13]. Additionally, a study performed on 65 children aged 10–16 with EDS demonstrated a high prevalence of a variety of sleep disorders including insomnia, circadian rhythm disorders, hypersomnia, and periodic limb movement disorder. Approximately 1/3 of the children were managed by behavioral sleep medicine [15]. A diagnosis of EDS and poor sleep independently contribute to a reduced QoL; however, there is a lack of current knowledge about overall sleep characteristics and potential association or interaction of these variables in these patients.

Within clinical practice, we note that many EDS patients complain of symptoms such as lack of good positioning for sleep due to pain, disordered sleep cycles, anxiety, depression, brain fog, memory loss, and concentration difficulties. All these symptoms can have contributing effects on the quality and length of sleep. Sleep characteristics have been heavily documented for disorders such as Parkinsonism [16] and Alzheimer’s disease [17], but little is known about these characteristics with EDS. In this study, we aim to characterize and uncover any significant underlying patterns within the sleep habits of patients with EDS.

## 2. Materials and Methods

### 2.1. Data Collection

An electronic survey entitled Characteristics of Sleep and Sleep Disorders in Individuals with Ehlers–Danlos Syndrome (Appendix A) was created and distributed electronically through the Ehlers–Danlos Society (www.ehlers-danlos.com accessed on 1 February 2022) to constituent members that have EDS. Inclusion criteria included being at least 18 years of age and having a diagnosis of EDS. The participation was voluntary, and patients were consented and had to select “Yes” to the first question, which acknowledged and agreed to the informed consent to enter the survey. All information was kept anonymous, and the data was gathered securely and protected through QuestionPro (www.questionpro.com accessed on 1 February 2022). The research project was given an exempt status by the University of Tennessee at Chattanooga Institutional Review Board (IRB # 21-133). The survey was opened on 1 October 2021 and stayed open until 1 February 2022, for a total of five months.

After collecting basic demographic information, questions were asked relating to the quantity and quality of sleep as well as sleep habits, fatigue, and strategies/interventions to improve sleep, for a total of 54 questions. The Participant-Reported Outcomes Measurement Information System (PROMIS) sleep scale was one component of the survey. This 27-question survey is a validated measure of overall sleep quality, disturbances, and satisfaction [18]. Digital fingerprinting, which encodes metadata from the device used to participate, was used to identify and potentially disqualify duplicate responses prior to data analysis.

### 2.2. Statistical Analysis

Basic descriptive statistics were first run to characterize the sample. Binary regression models were then run to measure the associations between selected variables such as age, gender, number of hours of sleep, use of sleep aids, and self-reporting of the number of days of poor mental or physical health in the last 30 days, to examine the potential existence of relationships. The number of days of poor mental or physical health reported by respondents were then grouped into either more or less than 14 days reported, which is a widely accepted definition of substantial level of impairment by the CDC. The confidence intervals for all analyses were set to 95%. All statistical analyses were performed utilizing the SPSS statistical package (IBM Version 29).

## 3. Results

### Study Population and Characteristics

Approximately 85% of the study population self-reported their EDS subtype was hypermobile EDS [19]. The gender distribution of the study population was skewed with females (*n* = 2216) accounting for 93.7% of study participants. The age ranges most represented in the study were ages 25–34 and 35–44, accounting for 23.3% and 26.2%, respectively. White persons accounted for most of the study population, with 93.1% (*n* = 2202) identifying as such. Regarding sleep quality, 39.1% (*n* = 924) of the study population reported sleeping between 6 and 7.9 h per night in the last week. Regarding time to sleep, 29.9% (*n* = 708) reported taking 31–60 min to fall asleep. Secondary diagnoses, particularly sleep apnea, could have also contributed to decreased quality of sleep and was present in 20.04% of respondents. Additionally, 36.38% of respondents reported that pain affected the quality of their sleep every day. A full characterization of the study population, separated by hEDS subtype, the other subtypes, and their sleep habits, can be found in Table 1.

Due to the categorization of the hours of sleep per day question in the survey and following the CDC recommendations of more than 7 h per night of sleep as being optimal, the analyses were run for patients reporting less than eight hours per day and less six hours per day. Binary logistic regression models revealed that sleeping fewer than eight hours per 24 h period was not a predictor for reporting more than 14 days of poor physical health (χ^2^(1) = 0.57, *p* = 0.450, *R*^2^ = 0.00), or poor mental health (χ^2^(1) = 0.78, *p* = 0.377, *R*^2^ = 0.00).

Binary logistic regressions were also run to determine if patients that reported sleeping less than 6 h had increased odds of reporting more than 14 days per month of poor physical health. Since increasing age can be a factor in experiencing more days of poor physical health, and the sample population was overwhelmingly female, the regression analysis was re-run, adjusting for age and gender to determine if these covariates affected the odds of experiencing more than 14 days of physical poor health per month. The overall model was significant (χ^2^(8) = 32.94, *p* < 0.001, R^2^ = 0.01), suggesting that these variables had a significant effect on the odds of reporting more than 14 days of poor physical health in the last 30 days. Regarding age and the odds of reporting more than 14 days of poor physical health, persons aged 65–74 had a 60% decrease in the odds (OR = 0.40, *p* < 0.001). Sleeping more than 6 h a day was also associated with a 32% decrease in the odds of reporting more than 14 days of poor physical health (OR = 0.68, *p* = 0.002).

Binary logistic regressions were run to determine if patients that reported sleeping less than 6 h, adjusting for age and gender, were more likely to report more than 14 days per month of poor mental health. The overall model was significant (χ^2^(8) = 89.25, *p* < 0.001, R^2^ = 0.03), suggesting that these variables had a significant effect on the odds of reporting more than 14 days of poor mental health. When examining the individual variables, all age groups except ages 25–34 were associated with decreased odds of reporting more than 14 days of poor mental health in a month, ranging from a 17 to 61% decrease. Sleeping more than 6 h was associated with a 26% decrease in odds of reporting more than 14 days of poor mental health (OR = 0.74, *p* = 0.002). While these results were significant for both models, the clinical relevance is uncertain due to the low R^2^ values. Table 2 and Table 3 presents the full results of these analyses.

Descriptive statistics were run to characterize the use of sleep aids in the study population. Frequencies and percentages were calculated for each of the sleep aids. The most frequently used sleep aids were positioning pillows or wedges, used by 76.32% of patients. Ear plugs were the most uncommon sleep aid utilized, with only 18.14% of patients reporting use. Table 4 presents the full results of the descriptives statistics.

A binary logistic regression was run to examine if the use of various sleep aids was a predictor for patients reporting more than 8 h of sleep in a 24 h period over the past week. The overall model was not significant, χ^2^(11) = 14.66, *p* = 0.198, *R*^2^ = 0.005, suggesting that the use of sleep aids did not significantly predict the odds of a person receiving more than 8 h of sleep in a 24 h period. Although the overall model was not significant for sleep aids affecting sleep length, the individual predictors were further examined. Patients who used a weighted blanket had 23% increased odds of sleeping more than 8 h in a 24 h period. The use of a white nose machine was also significant, but in an inverse direction (OR = 0.79, *p* = 0.018), indicating that the use of a white noise machine increased the odds of sleeping fewer than 8 h in a 24 h period by 21%. However, these results need to be interpreted cautiously as the overall model was not significant, and there may be confounding of some results, e.g., persons with existing sleep troubles are more likely to use sleep aids. Table 5 presents the full results of this analysis.

A binary logistic regression was run to examine if the use of various sleep aids was a predictor for patients reporting they had slept more than 6 h a night in the past week. The overall model was not significant, (χ^2^(11) = 3.33 *p* = 0.986, *R*^2^ = 0.001), suggesting that the use of sleep aids did not significantly predict the odds of a person receiving more than 6 h of sleep in a 24 h period. Although the overall model was not significant, the individual predictors underwent individual examination. None of the sleep aids when examined individually had an impact on the odds of a person sleeping more than 6 h. Table 6 presents the full results of this analysis.

## 4. Discussion

This study aimed to explore the sleep patterns and habits of individuals with EDS and how their sleep characteristics affect their overall well-being. In our study, 85.0% of respondents had hEDS, which is the most common type of EDS. This agrees with statistics from the Ehlers–Danlos Society stating that hEDS accounts for about 90% of all EDS cases [20]. Similarly, in our study, 93.7% of participants were females, also aligning with current prevalence research where females account for the vast majority of EDS cases [21]. The results of our study are therefore most applicable to females with hEDS. However, when discussing sleep disorders, it is important to acknowledge that gender disparities in sleep patterns exist in the healthy population, with women typically reporting poorer quality and more disrupted sleep [22]. More information is needed related to gender differences and sleep in people with EDS.

The recommended amount of sleep for adults is 7–9 h each day [6]. Our analysis showed sleep disturbance is highly prevalent within the EDS population, with up to 65.3% of participants reporting less than 8 h of sleep per night, and 26.2% with less than 6 h of sleep per night. This is slightly higher than the 23% of the general population reporting less than 6 h of sleep as published by the United States Centers for Disease Control and Prevention in 2016 [23]. That same study reported that 27.7% of the population reported 8 h of sleep per night as compared to 21% in our sample [23]. Additionally, regarding sleep latency, 67.5% of our respondents reported taking greater than 30 min to fall asleep and 37.6% taking greater than 60 min, which far exceeds sleep latency times of the general population, which is 11.7 min [24]. This decrease in sleep duration and increased sleep latency does not appear to be related to sleep apnea. In our sample, 20.04% had a diagnosis of sleep apnea, which aligns with current population values; however, no systematic assessment of comorbidities was conducted [25]. The experience of pain may also directly impact and be impacted by sleep. In our study, less than 4% of participants reported no pain or experiencing pain less than once per week while 52.31% reported pain every day. Short sleep duration has been associated with higher odds of chronic musculoskeletal pain [26]. The bidirectional link of pain and poor sleep must be addressed, as pain can also negatively impact sleep duration and quality, which can contribute to the perpetuation of a poor sleep cycle for EDS patients [27]. This data shows the high prevalence of sleep disorders among EDS patients. Difficulty falling asleep, coupled with shorter sleep durations, may contribute to the overall poor sleep quality reported, which may subsequently negatively impact QoL as it relates to mental and physical health.

A major focus of this study was the relationship between self-reported physical/mental health and sleep duration. This study found that there was not a significant relationship between patients reporting less than 8 h of sleep in a 24 h period and the likelihood of subsequently reporting poor physical and mental health days. However, there was a significant relationship between those sleeping less than 6 h with poor QoL as it relates to physical and mental health. The respondents who slept less than 6 h per night had increased odds of reporting more than 14 days of poor mental or physical health in the last 30 days. Interestingly, women in this study aged 65–74 actually had decreased odds of poor mental and physical health (>14 days per month). This might be due to length of time with hEDS and their adaptations in living with the disease. The findings of this study supports prior research that shows that improved health is associated with acquiring closer to the general recommendations for sleeping 7–9 h each night [28]. In otherwise healthy adults, let alone people with EDS, sleep disruption causes numerous detrimental effects on health including increased stress responsivity, pain, reduced QoL, emotional distress, mood disorders, and cognitive, memory, and performance deficits [29]. However, caution should be used in interpreting the clinical relevance of the results, as there may be confounding factors that influenced the results and the R^2^ values of the models are very low, so the variability in the dependent variable is poorly explained through the model.

As seen in Table 2, we were able to identify the impact of individual sleep aids on sleep duration. For many patients with sleep disorders, sleep aids become increasingly important. Many people in this study used sleep aids ranging from simple interventions such as an eye mask and ear plugs to more extensive interventions such as white noise machines and blackout curtains. The most common sleep aids were the use of positioning pillows (76.32%) and adjusting room temperature (63.42%). The use of a weighted blanket was shown to increase the odds of patients reporting more than 8 h of sleep whereas the use of a white noise machine significantly decreased the odds of patients reporting more than 8 h of sleep. However, the overall model was not significant, so caution should again be used for clinical interpretation. Medication use was also more common in the survey respondents. In our sample population, 41.40% of respondents used prescription sleep medication regularly compared to 8.4% of the general population [30]. Despite these interventions, only 34.7% of participants reported getting at least 8 h of sleep per night. This is not to say that sleeping aids are not beneficial as sleep quality and duration could have been poorer if these devices were not used. Additionally, many respondents may have felt that the aids offered some benefit, or they would not continue to regularly utilize them. Our results indicate that finding effective sleeping aids may be better determined on a case-by-case basis, as no intervention stood out as superior to the others. However, the various strategies listed in Table 4 can provide a starting point for physicians and offer suggestions to patients with EDS to help improve their sleep.

Our study findings may have clinical implications. It is known within the literature that EDS and sleep separately have profound impacts on psychological well-being [31,32]. With a high prevalence of psychologic disturbance and sleep disorders within the EDS population, incorporating comprehensive sleep assessment into the routine evaluation of EDS is integral. The role of the clinician is not only to treat obvious physical symptoms, but to help improve the QoL for patients. Physicians and other health care providers play a unique role in suggesting potential sleep interventions, which could assist patients in increasing sleep duration and quality, ultimately leading to an increased quality of life. Understanding the interplay between these factors can offer any physician an opportunity to provide significant benefit to this unique patient population.

## 5. Limitations

Within all studies, limitations do exist. This study relies on self-reported data, which may introduce bias, as participants may underestimate or overestimate their sleep patterns and health status. It is also possible that some respondents confused time spent in bed with time sleeping, which may have affected the results. Additionally, the survey asked for hours of sleep in a 24 h period but did not ask the respondents to specify how much, if any, naps accounted towards the overall daily sleep total. Other behavioral contributors to poor sleep, including meal times and screen use before bedtime, were not assessed, and limits the ability to evaluate how these behaviors may have contributed to sleep quality. The survey respondents self-reported their EDS diagnoses, meaning it is likely these diagnoses do not follow all the current 2017 guidelines of EDS classification. This may limit internal validity and generalizability of this study [1]. Also, certain patient characteristics and other medical conditions, which may impact sleeping habits, such as body mass index (BMI), menstruation status, psychiatric disorders, parenting responsibilities, etc., were not collected in this questionnaire. The sex and race/ethnicity makeup of the survey is also a limitation, as over 90% of survey participants identified as either white or female, which may potentially limit the generalizability of this study.

Selection bias is also a factor in this study, as the patients were voluntarily recruited using a website for EDS patients, which could skew the population to those that are more likely to report sleeping difficulties. This study also collected subjective measures without the ability to verify with objective clinical data, which should lead to caution in some data interpretation. Confounding variables may also limit some of the findings regarding sleep aids, as people who are experiencing poor sleep are more likely to use sleep aids. This study is also limited by its cross-sectional nature, which limits the ability to infer causality between sleep characteristics and QoL as defined by self-reported days of poor mental and physical health. In addition, the R^2^ values of the regression models are very low, indicating that much of the variability in the dependent variable is poorly explained. This can limit the clinical relevance of the findings, and while this could also be due to the cross-sectional nature of this study, further investigation is warranted.

## 6. Conclusions

This study highlights the complex interplay between sleep characteristics and habits in EDS patients. The results demonstrate an association between people with EDS and poorer sleep duration, increased sleep latency, and the frequent use of sleep aids including prescription sleep medication. While sleep quality is multi-factorial and more research needs to be performed in this area for people with EDS, our study provides preliminary evidence that the assessment of sleep and sleep hygiene may be important aspects to address in the management of EDS.

## Figures and Tables

**Table 1 medsci-13-00085-t001:** Study population and characteristics.

Demographic Variable	All	hEDS	Other EDS
	*n* = 2365	*n* = 2011	*n* = 347
**Gender**			
Male	80 (3.4%)	66 (3.3%)	14 (4.1%)
Female	2216 (93.7%)	1891 (94.1%)	323 (93.6%)
Prefer not to refer	60 (2.5%)	52 (2.6%)	8 (2.3%)
Missing	9 (0.4%)		
**Age**			
18–24	328 (13.9%)	282 (14.0%)	46 (13.3%)
25–34	551 (23.3%)	482 (24.0%)	68 (19.7%)
35–44	619 (26.2%)	535 (26.6%)	84 (24.3%)
45–54	426 (18.0%)	364 (18.1%)	61 (17.6%)
55–64	313 (13.2%)	253 (12.6%)	60 (17.3%)
65–74	109 (4.6%)	85 (4.2%)	24 (6.9%)
75+	13 (0.5%)	10 (0.5%)	3 (0.9%)
Missing	6 (0.3%)		
**Race**			
Caucasian	2202 (93.1%)	1888 (94.0%)	312 (90.4%)
Asian	23 (1.0%)	18 (0.9%)	5 (1.4%)
African American	7 (0.3%)	6 (0.3%)	1 (0.3%)
Other	95 (4.0%)	75 (3.7%)	20 (5.8%)
Prefer not to say	29 (1.2%)	22 (1.1%)	7 (2.0%)
Missing	9 (0.4%)		
**Sleep Quality Variable**			
**Average amount of sleep per 24 h period**			
0–5.9 h	620 (26.2%)	505 (25.1%)	115 (33.1%)
6–7.9 h	924 (39.1%)	799 (39.8%)	124 (35.7%)
8–9.9 h	497 (21.0%)	436 (21.7%)	60 (17.3%)
10–11.9 h	217 (9.2%)	187 (9.3%)	30 (8.6%)
12+ h	101 (4.3%)	83 (4.1%)	18 (5.2%)
Missing	6 (0.3%)		
**Sleep Latency**			
0–30 min	769 (32.5%)	670 (33.4%)	99 (28.8%)
31–60 min	708 (29.9%)	620 (30.9%)	87 (25.3%)
61–90 min	379 (16.0%)	314 (15.7%)	64 (18.6%)
91–120 min	254 (10.7%)	211 (10.5%)	43 (12.5%)
120+ min	242 (10.2%)	191 (9.5%)	51 (14.8%)
Missing	13 (0.5%)		

hEDS = hypermobile Ehlers–Danlos Syndrome.

**Table 2 medsci-13-00085-t002:** Results for binary logistic regression with people sleeping fewer than 6 h, grouped by age and gender, reporting more than 14 days of poor physical health in the last 30 days.

Variable	*B*	*SE*	*χ* ^2^	*p*	*OR*	95.00% CI
(Intercept)	1.27	0.30	18.09	<0.001	-	-
25–34 years old	0.04	0.18	0.04	0.835	1.04	[0.74, 1.46]
35–44 years old	−0.19	0.17	1.27	0.259	0.83	[0.59, 1.15]
45–54 years old	−0.15	0.19	0.68	0.411	0.86	[0.60, 1.24]
55–64 years old	−0.07	0.20	0.14	0.712	0.93	[0.62, 1.38]
65–74 years old	−0.93	0.25	14.15	<0.001	0.40	[0.24, 0.64]
75+ years old	−0.93	0.59	2.49	0.114	0.39	[0.12, 1.25]
Female gender	0.41	0.26	2.58	0.108	1.51	[0.91, 2.50]
Sleep > 6 h a day	−0.39	0.12	9.99	0.002	0.68	[0.53, 0.86]

Note. χ^2^(8) = 32.94; *p* < 0.001; McFadden *R*^2^ = 0.01.

**Table 3 medsci-13-00085-t003:** Results for binary logistic regression with people sleeping fewer than 6 h, grouped by age and gender, reporting more than 14 days of poor mental health in the last 30 days.

Variable	*B*	*SE*	*χ* ^2^	*p*	*OR*	95.00% CI
(Intercept)	0.67	0.27	6.39	0.011	-	-
25–34 years old	−0.13	0.15	0.77	0.380	0.88	[0.66, 1.17]
35–44 years old	−0.58	0.14	16.53	<0.001	0.56	[0.42, 0.74]
45–54 years old	−0.48	0.15	9.56	0.002	0.62	[0.46, 0.84]
55–64 years old	−0.95	0.17	32.34	<0.001	0.39	[0.28, 0.54]
65–74 years old	−1.44	0.24	34.83	<0.001	0.24	[0.15, 0.38]
75+ years old	−1.77	0.67	6.97	0.008	0.17	[0.05, 0.63]
Female gender	0.08	0.24	0.10	0.746	1.08	[0.68, 1.71]
Sleep > 6 h a day	−0.30	0.10	9.22	0.002	0.74	[0.61, 0.90]

Note. χ^2^(8) = 89.25; *p* < 0.001; McFadden *R*^2^ = 0.03.

**Table 4 medsci-13-00085-t004:** Frequency table for sleep aid use.

Variable	*n*	%
Use of a Weighted Blanket		
No	1785	75.48
Yes	580	24.52
Use of an Eye mask		
No	1936	81.86
Yes	429	18.14
Use of Ear Plugs		
No	2055	86.89
Yes	310	13.11
Use of a White Noise Machine		
No	1673	70.74
Yes	692	29.26
Modified Room Temperature		
No	865	36.58
Yes	1500	63.42
Use of Blackout Curtains/Light Barriers		
No	1199	50.70
Yes	1166	49.30
Use of Positioning Pillows or Wedges		
No	560	23.68
Yes	1805	76.32
Use of Mindfulness/Calming Phone Apps		
No	1766	74.67
Yes	599	25.33
Clothing		
No	1661	70.23
Yes	704	29.77
Use of a Prescription Sleep Medication		
No	1374	58.10
Yes	979	41.40
Use of an Over-The-Counter Sleep Medication		
No	1534	64.86
Yes	817	34.55

Note. Due to rounding errors, percentages may not equal 100%.

**Table 5 medsci-13-00085-t005:** Binary logistic regression results with the use of sleep aids predicting patients reporting sleeping more than 8 h per night in the last week.

Variable	*B*	*SE*	*Wald*	*p*	*OR*	95.00% CI
(Intercept)	0.72	0.11	43.13	<0.001	-	-
Weighted Blanket	0.21	0.10	4.14	0.042 *	1.23	[1.01, 1.50]
Eye Mask	0.20	0.12	3.05	0.081	1.25	[0.98, 1.54]
Ear Plugs	0.01	0.13	0.01	0.927	1.01	[0.78, 1.31]
White Noise Machine	−0.24	0.10	5.57	0.018 *	0.79	[0.65, 0.96]
Positioning Aids	0.12	0.11	1.24	0.266	1.13	[0.91, 1.40]
Lighting	−0.05	0.09	0.27	0.600	0.95	[0.79, 1.15]
Room Temperature	0.02	0.010	0.05	0.821	1.02	[0.84, 1.24]
Calming/Mindfulness Phone Apps	0.009	0.10	0.01	0.93	1.01	[0.82, 1.24]
Clothing	−0.05	0.10	0.28	0.595	0.95	[0.78, 1.15]
Over-the-Counter Medication	0.006	0.09	0.00	0.952	1.01	[0.84, 1.21]
Prescription Medication	−0.04	0.09	0.17	0.684	0.96	[0.81, 1.15]

Note. χ^2^(11) = 14.67; *p* = 0.198; McFadden *R*^2^ = 0.005. * = significant finding.

**Table 6 medsci-13-00085-t006:** Binary logistic regression results with the use of sleep aids predicting patients reporting sleeping more than 6 h per night in the last week.

Variable	*B*	*SE*	Wald	*p*	*OR*	95.00% CI
(Intercept)	1.10	0.12	88.11	<0.001	-	-
Weighted Blanket	−0.12	0.11	1.29	0.257	0.88	[0.71, 1.10]
Eye Mask	0.09	0.13	0.50	0.481	1.10	[0.85, 1.41]
Ear Plugs	−0.006	0.14	0.0	0.965	0.99	[0.75, 1.32]
White Noise Machine	0.04	0.11	0.12	0.727	1.04	[0.84, 1.28]
Positioning Aids	−0.02	0.12	0.02	0.898	0.99	[0.78, 1.24]
Lighting	−0.03	0.10	0.08	0.775	1.03	[0.84, 1.26]
Room Temperature	−0.03	0.11	0.10	0.751	0.97	[0.79, 1.19]
Calming/Mindfulness Phone Apps	−0.09	0.11	0.72	0.396	0.91	[0.73, 1.13]
Clothing	−0.02	0.11	0.05	0.827	0.98	[0.79, 1.21]
Over-the-Counter Medication	0.04	0.10	0.18	0.672	1.04	[0.86, 1.27]
Prescription Medication	−0.05	0.10	0.30	0.587	0.95	[0.78, 1.15]

Note. χ^2^(11) = 14.67; *p* = 0.198; McFadden *R*^2^ = 0.005.

## Data Availability

The raw data supporting the conclusions of this article will be made available by the authors on request.

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
