# Peer review of "Sleep Characteristics in Individuals with Ehlers-Danlos Syndrome"

_medsci, 2025, doi:10.3390/medsci13030085_

Round 1
Reviewer 1 Report
Comments and Suggestions for Authors
The authors investigated the sleep habits of patients with EDS using a specially developed questionaire including the validated PROMIS, which was distributed electronically within the Ehlers-Danlos-Society. The number of 2365 patients is enormous for this rare disease and represents certainly the largest database for EDS.
There are few comments:
1. EDS in sleep medicine is used for excessive daytime sleepiness. I recommend an explanation for this overlap to the reader.
2. Table 1 should have a legend for hEDS
3. Table 4: did patients choose room temperature up or down? Please clarify.
4. Did authors look for differences between patients that used medication regularly and those who did not? What type of hypnotic or over the counter medication had effect on sleep length? I think the questionnaire did not ask for specific medication. Since this impacts the results and conclusions, in depth interviews with selected patients could help to solve this problem.
5. P8 line 173: There should also be a discussion about those patients who slept >9 h which is larger than those who slept < 8 h
6. P8 line 177: a table with comorbid sleep disorders should be presented
7. P9 line 185: This major focus belongs into the introduction (primary goals)
8. P9 line 222: Since authors highlight the importance of improving Quol for patients they should present results
9. Discussion should include comparisons with the existing studies as listed in the introduction.
Considering the large amount of questions and the enormous population size, this study should come up with many more details. If this assumption is wrong it should be commented by the authors
Author Response
Thank you for your insight and valuable recommendations that have helped to improve the quality of our paper. Below is a list of comments and our actions:
- Table 1 should have a legend for hEDS.
- This was completed.
- Table 4: Did the patients choose from a room temperature:
- This was not asked in the questionnaire. However we changed the variable to more accurately reflect the ask.
- Did authors look for differences between patients that used medication regularly and those who did not? What type of hypnotic or over the counter medication had effect on sleep length? I think the questionnaire did not ask for specific medication. Since this impacts the results and conclusions, in depth interviews with selected patients could help to solve this problem.
- We did not look at the differences between medication use and sleep. This is a good suggestion for a post-hoc analysis.
- P8 line 173: There should also be a discussion about those patients who slept >9 h which is larger than those who slept < 8 h.
- The percentage of people who slept > 8 hours is less than the percentage of people who slept < 8 hours. Approximately 34 - 66%.
- P8 line 177: a table with comorbid sleep disorders should be presented
- We only have information on sleep apnea specifically, no other disorders
- P9 line 185: This major focus belongs into the introduction (primary goals)
- The Introduction was revised
- P9 line 222: Since authors highlight the importance of improving Quol for patients they should present results
- The Introduction was reviseDiscussion should include comparisons with the existing studies as listed in the introduction.
- Discussion should include comparisons with the existing studies as listed in the introduction. Considering the large amount of questions and the enormous population size, this study should come up with many more details. If this assumption is wrong, it should be commended by the authors
- The Discussion was revised and update
-
Major Comments
-Self-reported EDS diagnosis: While the large sample size is a strength, relying on self-reported diagnoses raises concerns about the accuracy and consistency of EDS classification. This is especially relevant given ongoing debates around diagnostic criteria for hypermobile EDS. The authors should discuss how this may affect both the internal validity of the study and the generalizability of the findings. –
Response: This was addressed in limitations
-Selection and recall bias: Recruiting participants through the Ehlers-Danlos Society website and using a voluntary survey approach likely introduces selection bias, potentially overrepresenting individuals with more severe symptoms or prominent sleep issues. This should be acknowledged as it may skew the sample toward those more motivated to report sleep problems.
Response: This was addressed in limitations
-Cross-sectional design and causality: The reported associations between short sleep duration and poorer self-rated health are noteworthy, but the cross-sectional nature of the data means causal conclusions cannot be drawn. The authors should adjust any causal language accordingly—phrases like “may warrant further investigation” or “is associated with” are more appropriate than definitive statements about impact or causation.
Response: This was addressed throughout in the rewording of the discussion and addition in limitations section.
Low explanatory power of regression models: The regression models yielded very low R² values (0.00–0.01), suggesting that sleep duration accounts for only a minimal proportion of the variance in physical or mental health outcomes. This limits the clinical relevance of the associations and should be clearly stated in the discussion.
Response: This was addressed in discussion/limitations section.
-Interpretation of sleep aid findings: The observed link between weighted blanket use and shorter sleep duration is likely confounded by indication—individuals experiencing more severe sleep problems may be more likely to use such aids. The authors should avoid causal interpretations and instead emphasize the need for tailored sleep interventions, acknowledging that such associations may reflect underlying symptom severity rather than treatment efficacy.
Response: This was addressed in the limitations section
Minor Comments and Language Corrections
Typographical errors:
Replace “white nose machine” by “white noise machine” throughout.
Response: This was addressed
Replace “weighted blanked”by “weighted blanket.”
Response: This was addressed
Clarity of terms:
“Use of clothing” as a sleep aid is vague; please specify whether this refers to compression garments, sleepwear, or other garments.
Response: we did not ask specifically about the type of clothing that was used as a sleep aid.
Grammar and usage:
Replace “sleeping < 8 hours” with “sleeping fewer than 8 hours.”
Response: This was addressed
Reword ambiguous sentences such as “EDS and poor sleep individually contribute...” to “EDS and poor sleep each independently contribute...” -
Response: This was addressed
Remove redundancies such as “respondents reported reporting” “respondents reported.”
Response: This was addressed
Specific Sections
Abstract
Ambiguous phrase: “Those who reported < 6 hours of sleep reported more poor mental and physical health days compared to those who slept < 8 hours.” The correct comparison should be between <6 h vs. ≥6 h, not <6 h vs. <8 h (which includes <6 h). This should be clarified.
Response: This was addressed by removing part of ambiguous statement.
Inconsistent phrasing: “Sleep aids were commonly used, included 41.40%...” “including 41.40%...”
Response: This was addressed
Unsupported conclusion: “...increased use of sleep aids including prescription sleep medication compared to the general population.” The study does not provide data directly comparing to the general population; this should be revised.
Addressed: A 2020 citation regarding prescription sleep medication use was added.
Introduction
Grammar typo: “These problems may results in...” “may result in...”
Response: This was addressed
Outdated prevalence data and missing nuance on hypermobile EDS diagnosis complexity should be updated and discussed.
Response: Citations updated
Overgeneralization: “EDS affects every organ system...” could be refined to specify some subtypes.-
Response: The introduction was revised to reflect this
Lack of clear hypothesis or objectives; a clearer statement is recommended.
Response: Completed.
Methods
Diagnosis is self-reported, introducing possible misclassification; discuss extensively.
Response: This was addressed in limitations.
Lack of control or reporting of comorbidities that affect sleep, such as psychiatric disorders.
Response: This was addressed in limitations
Missing details on survey length, number of questions, and duplication control.
Response: This was addressed
Consent process should be clarified.
Response: This was addressed
Justify use of linear models for discrete variables and acknowledge low explanatory power.
Response: This was addressed in stats analysis methodology and limitations sections.
Results
Interpretation of models and clinical relevance should be cautious due to low R².
Response: This was added to the discussion and limitations
Weighted blanket usage association with short sleep likely confounded; discuss reverse causality.
Response: This was addressed in discussion and limitations.
Consider exploring differences between EDS subtypes. –
Response: hEDS is most prevalent subtype and thus we focused on this as the vast majority reported a diagnosis of hEDS vs other subtypes, We would not have sufficient power to compare in other subtypes.
Discussion
Avoid overinterpretation of statistically significant but clinically limited findings.
Response: This was addressed by revisions in discussion section.
Explicitly discuss limitations regarding self-report diagnosis and causality direction (reverse causality).
Response: This was addressed in limitations
Tone down strong conclusions to more cautious phrasing.
Response: This was revised in the discussion section to reflect softer language.
Limitations
Add discussion on self-selection bias and reliance on subjective measures without objective sleep data.
Response: This was addressed
Note the absence of psychological variables mediating sleep-health relationship.
Response: This was addressed
Supplemental
Survey design – general observations:
The survey is overall well-structured and covers a broad range of sleep-related topics that are highly relevant for individuals with EDS. The use of a validated sleep instrument (PROMIS Sleep Disturbance) adds strength and standardization to the data collected. Additionally, the inclusion of items on sleep aids and subjective sleep quality offers valuable insights for future clinical applications.
There are some important areas that could have been addressed more thoroughly. The survey does not include key variables known to influence sleep, such as comorbid psychiatric or medical conditions (e.g., anxiety, depression, BMI), occupational factors, or behaviors like screen use before bed. While total sleep time is assessed, it’s unclear whether this includes daytime napping, which is particularly relevant in populations with chronic pain or fatigue. Also, while the sample includes a very high proportion of women, the survey does not ask about sex-specific factors that may affect sleep (e.g., parenting responsibilities, menstrual changes).
Finally, some questions could benefit from more precise wording (e.g., “use of clothing” as a sleep aid is vague), and the self-reported diagnosis of EDS, though practical, limits diagnostic certainty.
Response: Thank you and we agree!
Comments on the Quality of English Language
The manuscript would benefit from language editing.
Response: The document was proofread and language quality improved
- The Discussion was revised and update
Reviewer 2 Report
Comments and Suggestions for Authors
General Comments
This manuscript addresses a clinically relevant and understudied topic: the sleep characteristics and habits of individuals with Ehlers-Danlos Syndrome (EDS). The study benefits from a large sample size (N=2365), the use of a validated instrument (PROMIS Sleep Scale), and collaboration with the Ehlers-Danlos Society for participant recruitment. The findings are of interest to both clinicians and researchers working in this field. there are several important limitations related to study design and interpretation. For instance, the manuscript doesn’t examine whether sleep patterns vary by sex, age, or other known modifiers like comorbidities, psychiatric conditions, or body weight. Given that 93.7% of the sample are women, this raises concerns about representativeness and potential sex-based differences that remain unexplored. Similarly, while sleep quality typically declines with age, no statistical adjustments or stratified analyses by age were conducted. I strongly encourage the authors to revisit key analyses, either stratifying by sex and age or, at a minimum, including them as covariates in regression models.
If BMI data were not collected, this should be clearly acknowledged, since body weight is a key factor in conditions like obstructive sleep apnea.
Given that most participants were women, many likely within reproductive or perimenopausal age ranges, female-specific factors—such as menstrual cycle regularity and parenting responsibilities (particularly caring for young children)—could also meaningfully influence sleep. The absence of these variables limits interpretation of reported sleep duration and quality.
The racial and ethnic makeup of the sample is another limitation. With 93.1% of respondents identifying as White and no subgroup analyses reported, the findings may not generalize to more diverse populations. The authors should explicitly acknowledge this limitation and, if possible, consider exploring patterns within the smaller non-White subgroups.
There are also some notable omissions in the sleep-related variables assessed. Daytime napping, for example, wasn’t included in the survey—even though it can substantially affect total sleep duration and might serve as a coping mechanism for those with chronic pain or fatigue. Without accounting for naps, the prevalence of short sleep may be overestimated.
Other key contextual factors were also missing. Work-related variables—such as employment status, hours worked per week, and shift type—can significantly shape sleep patterns and are especially relevant for individuals with fatigue or physical limitations. These would be useful to explore in future studies.
Screen exposure before bedtime is another factor known to impact sleep onset and quality through its effects on melatonin and circadian rhythms. While the survey touched on the use of sleep-related phone apps, it didn’t assess overall screen use, leaving a gap in behavioral context.
Finally, sleep timing and pre-sleep behaviors—such as eating close to bedtime—are central to good sleep hygiene and circadian health. In populations like EDS, where gastrointestinal symptoms are common, these behaviors might have a particularly strong impact. Not including them limits the ability to fully understand behavioral contributors to poor sleep in this group.
Major Comments
-Self-reported EDS diagnosis: While the large sample size is a strength, relying on self-reported diagnoses raises concerns about the accuracy and consistency of EDS classification. This is especially relevant given ongoing debates around diagnostic criteria for hypermobile EDS. The authors should discuss how this may affect both the internal validity of the study and the generalizability of the findings.
-Selection and recall bias: Recruiting participants through the Ehlers-Danlos Society website and using a voluntary survey approach likely introduces selection bias, potentially overrepresenting individuals with more severe symptoms or prominent sleep issues. This should be acknowledged as it may skew the sample toward those more motivated to report sleep problems.
-Cross-sectional design and causality: The reported associations between short sleep duration and poorer self-rated health are noteworthy, but the cross-sectional nature of the data means causal conclusions cannot be drawn. The authors should adjust any causal language accordingly—phrases like “may warrant further investigation” or “is associated with” are more appropriate than definitive statements about impact or causation.
-Low explanatory power of regression models: The regression models yielded very low R² values (0.00–0.01), suggesting that sleep duration accounts for only a minimal proportion of the variance in physical or mental health outcomes. This limits the clinical relevance of the associations and should be clearly stated in the discussion.
-Interpretation of sleep aid findings: The observed link between weighted blanket use and shorter sleep duration is likely confounded by indication—individuals experiencing more severe sleep problems may be more likely to use such aids. The authors should avoid causal interpretations and instead emphasize the need for tailored sleep interventions, acknowledging that such associations may reflect underlying symptom severity rather than treatment efficacy.
Minor Comments and Language Corrections
Typographical errors:
Replace “white nose machine” by “white noise machine” throughout.
Replace “weighted blanked”by “weighted blanket.”
Clarity of terms:
“Use of clothing” as a sleep aid is vague; please specify whether this refers to compression garments, sleepwear, or other garments.
Grammar and usage:
Replace “sleeping < 8 hours” with “sleeping fewer than 8 hours.”
Reword ambiguous sentences such as “EDS and poor sleep individually contribute...” to “EDS and poor sleep each independently contribute...”
Remove redundancies such as “respondents reported reporting” “respondents reported.”
Specific Sections
Abstract
Ambiguous phrase: “Those who reported < 6 hours of sleep reported more poor mental and physical health days compared to those who slept < 8 hours.” The correct comparison should be between <6 h vs. ≥6 h, not <6 h vs. <8 h (which includes <6 h). This should be clarified.
Inconsistent phrasing: “Sleep aids were commonly used, included 41.40%...” “including 41.40%...”
Unsupported conclusion: “...increased use of sleep aids including prescription sleep medication compared to the general population.” The study does not provide data directly comparing to the general population; this should be revised.
Introduction
Grammar typo: “These problems may results in...” “may result in...”
Outdated prevalence data and missing nuance on hypermobile EDS diagnosis complexity should be updated and discussed.
Overgeneralization: “EDS affects every organ system...” could be refined to specify some subtypes.
Lack of clear hypothesis or objectives; a clearer statement is recommended.
Methods
Diagnosis is self-reported, introducing possible misclassification; discuss extensively.
Lack of control or reporting of comorbidities that affect sleep, such as psychiatric disorders.
Missing details on survey length, number of questions, and duplication control.
Consent process should be clarified.
Justify use of linear models for discrete variables and acknowledge low explanatory power.
Results
Interpretation of models and clinical relevance should be cautious due to low R².
Weighted blanket usage association with short sleep likely confounded; discuss reverse causality.
Tables 5 and 6 could be condensed if not contributing significantly.
Consider exploring differences between EDS subtypes.
Discussion
Avoid overinterpretation of statistically significant but clinically limited findings.
Explicitly discuss limitations regarding self-report diagnosis and causality direction (reverse causality).
Tone down strong conclusions to more cautious phrasing.
Limitations
Add discussion on self-selection bias and reliance on subjective measures without objective sleep data.
Note the absence of psychological variables mediating sleep-health relationship.
Supplemental
Survey design – general observations:
The survey is overall well-structured and covers a broad range of sleep-related topics that are highly relevant for individuals with EDS. The use of a validated sleep instrument (PROMIS Sleep Disturbance) adds strength and standardization to the data collected. Additionally, the inclusion of items on sleep aids and subjective sleep quality offers valuable insights for future clinical applications.
There are some important areas that could have been addressed more thoroughly. The survey does not include key variables known to influence sleep, such as comorbid psychiatric or medical conditions (e.g., anxiety, depression, BMI), occupational factors, or behaviors like screen use before bed. While total sleep time is assessed, it’s unclear whether this includes daytime napping, which is particularly relevant in populations with chronic pain or fatigue. Also, while the sample includes a very high proportion of women, the survey does not ask about sex-specific factors that may affect sleep (e.g., parenting responsibilities, menstrual changes).
Finally, some questions could benefit from more precise wording (e.g., “use of clothing” as a sleep aid is vague), and the self-reported diagnosis of EDS, though practical, limits diagnostic certainty.
Comments on the Quality of English Language
The manuscript would benefit from language editing.
Author Response
Thank you for the valuable feedback to improve the quality of our manuscript. Below is a response to your comments:
This manuscript addresses a clinically relevant and understudied topic: the sleep characteristics and habits of individuals with Ehlers-Danlos Syndrome (EDS). The study benefits from a large sample size (N=2365), the use of a validated instrument (PROMIS Sleep Scale), and collaboration with the Ehlers-Danlos Society for participant recruitment. The findings are of interest to both clinicians and researchers working in this field. there are several important limitations related to study design and interpretation. For instance, the manuscript doesn’t examine whether sleep patterns vary by sex, age, or other known modifiers like comorbidities, psychiatric conditions, or body weight. Given that 93.7% of the sample are women, this raises concerns about representativeness and potential sex-based differences that remain unexplored. Similarly, while sleep quality typically declines with age, no statistical adjustments or stratified analyses by age were conducted. I strongly encourage the authors to revisit key analyses, either stratifying by sex and age or, at a minimum, including them as covariates in regression models.
Response: A post hoc analyses performed showed that age did not have an impact on the regression models
If BMI data were not collected, this should be clearly acknowledged, since body weight is a key factor in conditions like obstructive sleep apnea.
Response: This was addressed in limitations
Given that most participants were women, many likely within reproductive or perimenopausal age ranges, female-specific factors—such as menstrual cycle regularity and parenting responsibilities (particularly caring for young children)—could also meaningfully influence sleep. The absence of these variables limits interpretation of reported sleep duration and quality.
Response: This was addressed in limitations section
The racial and ethnic makeup of the sample is another limitation. With 93.1% of respondents identifying as White and no subgroup analyses reported, the findings may not generalize to more diverse populations. The authors should explicitly acknowledge this limitation and, if possible, consider exploring patterns within the smaller non-White subgroups.
Response: This was addressed in the limitations section.
There are also some notable omissions in the sleep-related variables assessed. Daytime napping, for example, wasn’t included in the survey—even though it can substantially affect total sleep duration and might serve as a coping mechanism for those with chronic pain or fatigue. Without accounting for naps, the prevalence of short sleep may be overestimated.
Response: Napping is included in the overall sleep within the 24 hour period, but a specific question around naps was not asked.
Other key contextual factors were also missing. Work-related variables—such as employment status, hours worked per week, and shift type—can significantly shape sleep patterns and are especially relevant for individuals with fatigue or physical limitations.
Response: These would be useful to explore in future studies. – We agree!
Screen exposure before bedtime is another factor known to impact sleep onset and quality through its effects on melatonin and circadian rhythms. While the survey touched on the use of sleep-related phone apps, it didn’t assess overall screen use, leaving a gap in behavioral context.
Response: Good point- we did not ask this question
Finally, sleep timing and pre-sleep behaviors—such as eating close to bedtime—are central to good sleep hygiene and circadian health. In populations like EDS, where gastrointestinal symptoms are common, these behaviors might have a particularly strong impact. Not including them limits the ability to fully understand behavioral contributors to poor sleep in this group.
Response: We agree- we did not ask this question.
Major Comments
-Self-reported EDS diagnosis: While the large sample size is a strength, relying on self-reported diagnoses raises concerns about the accuracy and consistency of EDS classification. This is especially relevant given ongoing debates around diagnostic criteria for hypermobile EDS. The authors should discuss how this may affect both the internal validity of the study and the generalizability of the findings. –
Response: This was addressed in limitations
-Selection and recall bias: Recruiting participants through the Ehlers-Danlos Society website and using a voluntary survey approach likely introduces selection bias, potentially overrepresenting individuals with more severe symptoms or prominent sleep issues. This should be acknowledged as it may skew the sample toward those more motivated to report sleep problems.
Response: This was addressed in limitations
-Cross-sectional design and causality: The reported associations between short sleep duration and poorer self-rated health are noteworthy, but the cross-sectional nature of the data means causal conclusions cannot be drawn. The authors should adjust any causal language accordingly—phrases like “may warrant further investigation” or “is associated with” are more appropriate than definitive statements about impact or causation.
Response: This was addressed throughout in the rewording of the discussion and addition in limitations section.
Low explanatory power of regression models: The regression models yielded very low R² values (0.00–0.01), suggesting that sleep duration accounts for only a minimal proportion of the variance in physical or mental health outcomes. This limits the clinical relevance of the associations and should be clearly stated in the discussion.
Response: This was addressed in discussion/limitations section.
-Interpretation of sleep aid findings: The observed link between weighted blanket use and shorter sleep duration is likely confounded by indication—individuals experiencing more severe sleep problems may be more likely to use such aids. The authors should avoid causal interpretations and instead emphasize the need for tailored sleep interventions, acknowledging that such associations may reflect underlying symptom severity rather than treatment efficacy.
Response: This was addressed in the limitations section
Minor Comments and Language Corrections
Typographical errors:
Replace “white nose machine” by “white noise machine” throughout.
Response: This was addressed
Replace “weighted blanked”by “weighted blanket.”
Response: This was addressed
Clarity of terms:
“Use of clothing” as a sleep aid is vague; please specify whether this refers to compression garments, sleepwear, or other garments.
Response: we did not ask specifically about the type of clothing that was used as a sleep aid.
Grammar and usage:
Replace “sleeping < 8 hours” with “sleeping fewer than 8 hours.”
Response: This was addressed
Reword ambiguous sentences such as “EDS and poor sleep individually contribute...” to “EDS and poor sleep each independently contribute...” -
Response: This was addressed
Remove redundancies such as “respondents reported reporting” “respondents reported.”
Response: This was addressed
Specific Sections
Abstract
Ambiguous phrase: “Those who reported < 6 hours of sleep reported more poor mental and physical health days compared to those who slept < 8 hours.” The correct comparison should be between <6 h vs. ≥6 h, not <6 h vs. <8 h (which includes <6 h). This should be clarified.
Response: This was addressed by removing part of ambiguous statement.
Inconsistent phrasing: “Sleep aids were commonly used, included 41.40%...” “including 41.40%...”
Response: This was addressed
Unsupported conclusion: “...increased use of sleep aids including prescription sleep medication compared to the general population.” The study does not provide data directly comparing to the general population; this should be revised.
Addressed: A 2020 citation regarding prescription sleep medication use was added.
Introduction
Grammar typo: “These problems may results in...” “may result in...”
Response: This was addressed
Outdated prevalence data and missing nuance on hypermobile EDS diagnosis complexity should be updated and discussed.
Response: Citations updated
Overgeneralization: “EDS affects every organ system...” could be refined to specify some subtypes.-
Response: The introduction was revised to reflect this
Lack of clear hypothesis or objectives; a clearer statement is recommended.
Response: Completed.
Methods
Diagnosis is self-reported, introducing possible misclassification; discuss extensively.
Response: This was addressed in limitations.
Lack of control or reporting of comorbidities that affect sleep, such as psychiatric disorders.
Response: This was addressed in limitations
Missing details on survey length, number of questions, and duplication control.
Response: This was addressed
Consent process should be clarified.
Response: This was addressed
Justify use of linear models for discrete variables and acknowledge low explanatory power.
Response: This was addressed in stats analysis methodology and limitations sections.
Results
Interpretation of models and clinical relevance should be cautious due to low R².
Response: This was added to the discussion and limitations
Weighted blanket usage association with short sleep likely confounded; discuss reverse causality.
Response: This was addressed in discussion and limitations.
Consider exploring differences between EDS subtypes. –
Response: hEDS is most prevalent subtype and thus we focused on this as the vast majority reported a diagnosis of hEDS vs other subtypes, We would not have sufficient power to compare in other subtypes.
Discussion
Avoid overinterpretation of statistically significant but clinically limited findings.
Response: This was addressed by revisions in discussion section.
Explicitly discuss limitations regarding self-report diagnosis and causality direction (reverse causality).
Response: This was addressed in limitations
Tone down strong conclusions to more cautious phrasing.
Response: This was revised in the discussion section to reflect softer language.
Limitations
Add discussion on self-selection bias and reliance on subjective measures without objective sleep data.
Response: This was addressed
Note the absence of psychological variables mediating sleep-health relationship.
Response: This was addressed
Supplemental
Survey design – general observations:
The survey is overall well-structured and covers a broad range of sleep-related topics that are highly relevant for individuals with EDS. The use of a validated sleep instrument (PROMIS Sleep Disturbance) adds strength and standardization to the data collected. Additionally, the inclusion of items on sleep aids and subjective sleep quality offers valuable insights for future clinical applications.
There are some important areas that could have been addressed more thoroughly. The survey does not include key variables known to influence sleep, such as comorbid psychiatric or medical conditions (e.g., anxiety, depression, BMI), occupational factors, or behaviors like screen use before bed. While total sleep time is assessed, it’s unclear whether this includes daytime napping, which is particularly relevant in populations with chronic pain or fatigue. Also, while the sample includes a very high proportion of women, the survey does not ask about sex-specific factors that may affect sleep (e.g., parenting responsibilities, menstrual changes).
Finally, some questions could benefit from more precise wording (e.g., “use of clothing” as a sleep aid is vague), and the self-reported diagnosis of EDS, though practical, limits diagnostic certainty.
Response: Thank you and we agree!
Comments on the Quality of English Language
The manuscript would benefit from language editing.
Response: The document was proofread and language quality improved
Round 2
Reviewer 1 Report
Comments and Suggestions for Authors
Table 3: <6h
P13: "When discussing sleep disorders, it is important to acknowledge gender disparities in sleep patterns [22]" Authors should discuss the impact of gender in EDS specifically for this study and not make a general comment.
p13, 2nd paragraph: "The bidirec-tional link of pain and poor sleep must be addressed, as pain can also negatively impact sleep duration and quality, which can contribute to the perpetuation of a poor sleep cy-cle for EDS patients [27]" authors should address this aspect for EDS and not repeat reviewers comments
p13: "ImprovedThis aligns with prior research that shows that improved health is associated with acquiring closer....." Please check the meaning of the sentence
p13: "The R2 values of the models are very low, so the variability of the dependent variable is poorly explained through the model and caution should be used in interpreting the clin-ical relevance of the results" Authors should break this statement down to the results of their study!
Limitations: Authors refer to the psychological problems of EDS patients, but have not assessed them. This should be mentioned.
Author Response
Thank you for your time, expertise and recommendations to help improve the quality of our paper. Below are our responses to your comments.
Reviewer 1
Table 3: <6h
Response: This change was made
P13: "When discussing sleep disorders, it is important to acknowledge gender disparities in sleep patterns [22]" Authors should discuss the impact of gender in EDS specifically for this study and not make a general comment.
Response: This part of the discussion was changed to: The results of our study are therefore most applicable to females with hEDS. However, when discussing sleep disorders, it is important to acknowledge that gender disparities in sleep patterns exist in the healthy population with women typically reporting poorer quality and more disrupted sleep [22]. More information is needed related to gender differences and sleep in people with EDS.
p13, 2nd paragraph: "The bidirec-tional link of pain and poor sleep must be addressed, as pain can also negatively impact sleep duration and quality, which can contribute to the perpetuation of a poor sleep cy-cle for EDS patients [27]" authors should address this aspect for EDS and not repeat reviewers comments
p13: "ImprovedThis aligns with prior research that shows that improved health is associated with acquiring closer....." Please check the meaning of the sentence
Response: This paragraph was re-written to help with clarity and meaning: There was not a relationship between patients reporting less than 8 hours of sleep per night and the likelihood of subsequently reporting poor physical and mental health days over a week period. There was a significant relationship between those sleeping less than 6 hours with poor QoL as it relates to physical and mental health. The respondents who slept less than 6 hours per night reported having 1.92 more days of poor physical health and 1.45 more days of poor mental health in the preceding week compared to those who slept more than 6 hours. This aligns with prior research that shows that improved health is associated with acquiring closer to the general recommendations for sleeping 7 – 9 hours each night [28]. The R2 values of our models are very low, so the variability of the dependent variable is poorly explained through the model and caution should be used in interpreting the clinical relevance of the results.
p13: "The R2 values of the models are very low, so the variability of the dependent variable is poorly explained through the model and caution should be used in interpreting the clin-ical relevance of the results" Authors should break this statement down to the results of their study!
Limitations: Authors refer to the psychological problems of EDS patients, but have not assessed them. This should be mentioned.
Response: This sentence was added to the limitations section: which can limit internal validity and generalizability of the study.[1] Also, certain patient characteristics and other medical conditions, which may impact sleeping habits, such as Body Mass Index (BMI), menstration status, psychiatric disorders, parenting responsibilities, etc., were not collected in this questionnaire.
Reviewer 2 Report
Comments and Suggestions for Authors
Revision 2
The authors have addressed many of the initial concerns, including language clarity, acknowledgment of major limitations, and typographical errors. The discussion now adopts a more cautious tone, and the limitations section has been expanded meaningfully. These changes improve the transparency and readability of the work. However, several key methodological and interpretive issues remain that limit the strength of the conclusions and should be addressed in a further revision.
Outstanding Issues and Required Revisions
Lack of adjustment for age and sex in regression models
While the authors state that a post hoc analysis showed no effect of age, no results from such analysis are reported, and neither age nor sex were included as covariates in the regression models. Given that 93.7% of the sample were women, and both age and sex are known to influence sleep patterns, failing to account for them limits the generalizability and internal validity of the findings.
I recommend include these covariates in the models or present the post hoc analyses and justify why adjustment was deemed unnecessary.
Use of linear regression for discrete/count outcomes
Linear regression was applied to outcomes such as 'number of poor physical/mental health days,' which are discrete and potentially non-normally distributed. This violates core assumptions of linear models. A more appropriate approach would involve Poisson or negative binomial regression, depending on data distribution.
In addition, there seems to be some confusion in the description of the regression methods. The manuscript states that “logistic regressions were performed for dichotomous independent variables,” when in fact logistic regression is used when the dependent variable is dichotomous. It would help to revise that description for clarity. Also, while linear regression was used with categorical predictors, there’s no explanation of how those predictors were handled—whether they were dummy coded, what reference categories were used, etc. Explaining these details would make the statistical approach easier to follow and evaluate.
There are still some important issues with the analysis that go beyond the choice of model. For example, no covariates such as age or sex were included, even though these are known to affect sleep outcomes and the sample is overwhelmingly female. Not adjusting for them weakens the internal validity of the results. Including these covariates—and being transparent about how categorical predictors were modeled—would significantly improve the overall analysis.
Although R² values are now acknowledged, some interpretations (e.g., regarding sleep aids or clinical impact of sleep duration) remain too strong considering that the models explain virtually no variance (R² = 0.00–0.01).
Emphasize the exploratory nature of findings and avoid implying clinical relevance where explanatory power is minimal.Low explanatory power of regression models
Interpretation of sleep aid use
The association between weighted blanket use and shorter sleep duration, while statistically significant, is likely confounded by indication—individuals with more severe sleep disturbances may be more likely to use such aids. The manuscript mentions this possibility but still frames the findings as suggestive of clinical impact.
Further temper conclusions around sleep aids and clearly acknowledge the possibility of reverse causality or unmeasured confounding.
Ambiguity around total sleep time and naps
The survey asked about total sleep time in 24 hours but did not specify whether this refers exclusively to nocturnal sleep or includes naps. Since the timing and structure of sleep (night vs. day) differ physiologically, failing to distinguish between them limits interpretability.
This should be clearly acknowledged as a limitation of the study.
Omission of key behavioral and contextual variables
Variables such as occupational status, screen use before bedtime, and meal timing were not assessed, all of which can significantly influence sleep. The absence of these factors reduces the ability to evaluate modifiable behavioral contributors to poor sleep.
Expand the limitations section to explicitly include the absence of these variables.
Confusing description of regression methods
The statistical methods section states that 'logistic regressions were performed for dichotomous independent variables,' which is incorrect—logistic regression is used when the dependent variable is dichotomous. Additionally, while linear regression was used for models with categorical predictors, there is no explanation of how those categorical variables were handled (e.g., dummy coding, reference groups).
Correct the regression terminology and briefly describe how categorical predictors were entered into the models to enhance transparency.
Claims about medical contributors to sleep problems
The discussion suggests that poor sleep 'does not appear to be related to other medical conditions,' yet no systematic assessment of comorbidities (psychiatric or physical) was conducted.
This statement should be revised or removed, as it is not supported by the data collected.
Suggested Additions to Limitations Section
Additionally, this study did not include several variables known to influence sleep quality and duration, such as occupational demands, screen exposure before bedtime, and timing of meals. Total sleep time was assessed over a 24-hour period, but the survey did not specify whether this included daytime naps or how sleep episodes were distributed throughout the day. This ambiguity may limit interpretability, as nocturnal and diurnal sleep differ in quality and physiological function. These gaps should be considered when interpreting the findings and their generalizability.
Final Recommendation
With these additional clarifications and corrections, the manuscript would present a stronger and more methodologically sound contribution to the literature on sleep and EDS. The topic is important and underexplored, but conclusions should be firmly grounded in the data and appropriately contextualized.
Author Response
Thank you for your time, expertise, and recommendations to help improve the quality of our paper. Below are the responses to the requested revisions.
Reviewer 2
The authors have addressed many of the initial concerns, including language clarity, acknowledgment of major limitations, and typographical errors. The discussion now adopts a more cautious tone, and the limitations section has been expanded meaningfully. These changes improve the transparency and readability of the work. However, several key methodological and interpretive issues remain that limit the strength of the conclusions and should be addressed in a further revision.
Outstanding Issues and Required Revisions
Lack of adjustment for age and sex in regression models
While the authors state that a post hoc analysis showed no effect of age, no results from such analysis are reported, and neither age nor sex were included as covariates in the regression models. Given that 93.7% of the sample were women, and both age and sex are known to influence sleep patterns, failing to account for them limits the generalizability and internal validity of the findings.
I recommend include these covariates in the models or present the post hoc analyses and justify why adjustment was deemed unnecessary.
Response: The analyses were redone and included the use of age and gender.
Use of linear regression for discrete/count outcomes
Linear regression was applied to outcomes such as 'number of poor physical/mental health days,' which are discrete and potentially non-normally distributed. This violates core assumptions of linear models. A more appropriate approach would involve Poisson or negative binomial regression, depending on data distribution.
Response: The analyses were redone to better reflect the data distribution and type
In addition, there seems to be some confusion in the description of the regression methods. The manuscript states that “logistic regressions were performed for dichotomous independent variables,” when in fact logistic regression is used when the dependent variable is dichotomous. It would help to revise that description for clarity. Also, while linear regression was used with categorical predictors, there’s no explanation of how those predictors were handled—whether they were dummy coded, what reference categories were used, etc. Explaining these details would make the statistical approach easier to follow and evaluate.
Response: These changes have been made
There are still some important issues with the analysis that go beyond the choice of model. For example, no covariates such as age or sex were included, even though these are known to affect sleep outcomes and the sample is overwhelmingly female. Not adjusting for them weakens the internal validity of the results. Including these covariates—and being transparent about how categorical predictors were modeled—would significantly improve the overall analysis.
Response: The analyses were redone and included the use of age and gender.
Although R² values are now acknowledged, some interpretations (e.g., regarding sleep aids or clinical impact of sleep duration) remain too strong considering that the models explain virtually no variance (R² = 0.00–0.01).
Emphasize the exploratory nature of findings and avoid implying clinical relevance where explanatory power is minimal.Low explanatory power of regression models
Response: The following sentence was added into the results section help clarify the findings: . While these results were significant, the clinical relevance is uncertain due to the low R2 values.
Interpretation of sleep aid use
The association between weighted blanket use and shorter sleep duration, while statistically significant, is likely confounded by indication—individuals with more severe sleep disturbances may be more likely to use such aids. The manuscript mentions this possibility but still frames the findings as suggestive of clinical impact.
Response: The following sentence was added to the results: These results need to be interpreted cautiously as there may be confounding variables influencing the results. For instance, those with greater sleeper disturbance may be more likely to use these sleeping aids
Further temper conclusions around sleep aids and clearly acknowledge the possibility of reverse causality or unmeasured confounding.
Response: The following sentence was added into the discussion: This aligns with prior research that shows that improved health is associated with acquiring closer to the general recommendations for sleeping 7 – 9 hours each night [28]. However, there may be confounding variables that the influenced the results and the R2 values of our models are very low, so the variability of the dependent variable is poorly explained through the model and caution should be used in interpreting the clinical relevance of the results
Ambiguity around total sleep time and naps
The survey asked about total sleep time in 24 hours but did not specify whether this refers exclusively to nocturnal sleep or includes naps. Since the timing and structure of sleep (night vs. day) differ physiologically, failing to distinguish between them limits interpretability.
This should be clearly acknowledged as a limitation of the study.
Response: The following sentence was added to limitations: It is also possible, that some respondents confused time spent in bed with time sleeping which may have affected our results. Additionally, the survey asked for hours of sleep at night and did not specify naps or sleeping during the day
Omission of key behavioral and contextual variables
Variables such as occupational status, screen use before bedtime, and meal timing were not assessed, all of which can significantly influence sleep. The absence of these factors reduces the ability to evaluate modifiable behavioral contributors to poor sleep.
Expand the limitations section to explicitly include the absence of these variables.
The following sentence was added to the limitations: Other behavioral contributors to poor sleep including meal times and screen use before bedtime were not assessed and limits our ability to evaluate how these behaviors may have contributed to poor sleep.
Confusing description of regression methods
The statistical methods section states that 'logistic regressions were performed for dichotomous independent variables,' which is incorrect—logistic regression is used when the dependent variable is dichotomous. Additionally, while linear regression was used for models with categorical predictors, there is no explanation of how those categorical variables were handled (e.g., dummy coding, reference groups).
Response: The analyses were redone and included the use of age and gender.
Correct the regression terminology and briefly describe how categorical predictors were entered into the models to enhance transparency.
Response: The analyses were redone and better reflect the distribution of the data.
Claims about medical contributors to sleep problems
The discussion suggests that poor sleep 'does not appear to be related to other medical conditions,' yet no systematic assessment of comorbidities (psychiatric or physical) was conducted.
This statement should be revised or removed, as it is not supported by the data collected.
Response: The following sentence was added to the discussion: This decrease in sleep duration and increased sleep latency does not appear to be related to sleep apnea. In our sample, 20.04% had a diagnosis of sleep apnea, which aligns with current population values; however, no systematic assessment of comorbidities was conducted [25].
Suggested Additions to Limitations Section
Additionally, this study did not include several variables known to influence sleep quality and duration, such as occupational demands, screen exposure before bedtime, and timing of meals. Total sleep time was assessed over a 24-hour period, but the survey did not specify whether this included daytime naps or how sleep episodes were distributed throughout the day. This ambiguity may limit interpretability, as nocturnal and diurnal sleep differ in quality and physiological function. These gaps should be considered when interpreting the findings and their generalizability.
Response: See the responses above.
Final Recommendation
With these additional clarifications and corrections, the manuscript would present a stronger and more methodologically sound contribution to the literature on sleep and EDS. The topic is important and underexplored, but conclusions should be firmly grounded in the data and appropriately contextualized.